# An Operational Capacity Assessment Method for an Urban Low-Altitude Unmanned Aerial Vehicle Logistics Route Network

Jia Yi [1,2], Honghai Zhang [1,2,*], Fei Wang [1,2], Changyuan Ning [1,2], Hao Liu [2,3] and Gang Zhong [1,2]

1   College of Civil Aviation, Nanjing University of Aeronautics and Astronautics, Nanjing 211106, China; yijia1106@nuaa.edu.cn (J.Y.); wangfei07@nuaa.edu.cn (F.W.); ningchangyuan@nuaa.edu.cn (C.N.)
2   National Key Laboratory of Air Traffic Flow Management, Nanjing University of Aeronautics and Astronautics, No. 29 General Avenue, Nanjing 211106, China
3   College of Mathematics, Nanjing University of Aeronautics and Astronautics, Nanjing 211106, China
*   Correspondence: honghaizhang@nuaa.edu.cn; Tel.: +86-13914766151

**Abstract:** The Federal Aviation Administration introduced the concept of urban air mobility (UAM), a new three-dimensional transport system that operates with a fusion of manned/unmanned aerial vehicles on an urban or intercity scale. The rapid development of UAM has brought innovation and dynamism to many industries, especially in the field of logistics. Various types of unmanned aerial vehicles (UAVs) for use in transport logistics are being designed and produced. UAV logistics refers to the use of UAVs, usually carrying goods and parcels, to achieve route planning, identify risk perception, facilitate parcel delivery, and carry out other functions. This research provides a method for assessing the operational capacity of a UAV logistics route network. The concept of "logistics UAV route network operation capacity" is defined, and a bi-objective optimization model for assessing the route network's operating capacity is developed. The first objective is to maximize the number of UAV logistics delivery plans that can be executed in a fixed operation time. The second objective is to minimize the total operational impedance value in a fixed operation time. To solve the bi-objective optimization model, the Non-dominated Sorting Genetic Algorithm-II (NSGA-II) is utilized. A UAV logistics route network with 62 nodes is developed to assess the rationale and validity of the proposed concept. The experiments show that with an increase in operation time, the route network's optimal operational capacity gradually increases, the convergence speed of the algorithm slows down, and the optimization magnitude gradually reduces. Two key parameters—operational safety interval and flight speed—are further analyzed in the experiments. According to the experiments, as the safety interval increases, the route network's average operational capacity steadily diminishes, as does its sensitivity to the safety interval. The average operational capacity steadily rose with the rise in flight speed, especially when the UAV logistics flight speed was between 10 m/s and 10.5 m/s. In that range, the operational capacity of the route network was substantially impacted by the flight speed.

**Keywords:** air traffic management; airspace management; capacity assessment; urban air mobility; UAV logistics

## 1. Introduction

The increasing urban population has caused traffic congestion and environmental pollution to become increasingly serious, especially in mega-cities such as New York, Shanghai, and Tokyo. Traditional urban transport and facilities are becoming increasingly inadequate to meet the growing demand. In recent years, with the continuous improvements in science and technology, the development of urban transport has trended toward green, intelligent, and sustainable features. In 2018, the Federal Aviation Administration (FAA) proposed the urban air mobility (UAM) concept, based on the small aircraft transportation system

(SATS) [1,2]. The development of UAM has been considered as an innovative solution to alleviate urban ground traffic congestion and improve transportation capacity.

Vertical takeoff and landing aircraft, such as electric vertical takeoff and landing (eVTOL) aircraft, rotor unmanned aerial vehicles (rotor UAVs), and helicopters, will be the leading examples of UAM. For specific application scenarios, different configurations of vertical takeoff and landing aircraft will be combined to operate in low-altitude urban airspace. In particular, the demand for UAV logistics is strong, and the market is vast. UAV logistics is a new type of transport that uses UAVs for package delivery.

Amazon Prime Air, Flytrex, UPS Flight Forward, and other services have begun to develop UAV logistics projects, one after another. Zipline has become a globally known company for scaling UAV logistics operations. Unlike companies such as Amazon, Zipline focuses on the delivery of medical supplies, especially in East Africa. To date, Zipline has successfully delivered blood packs to 21 hospitals within 75 km in Rwanda [3]. Meanwhile, DHL Express has partnered with EHang Command to propose a fully automated UAV logistics solution for the last mile of logistics delivery in China [4]. The MarketsandMarkets company predicts that the market for the use of UVA logistics in transportation will reach USD 17 billion by 2030 [5].

## 2. Related Works

(1)    Application of UAVs in logistics

UAVs were initially invented and produced as military products because of their flexibility, economy, and unmanned facility. The application of UAVs gradually expanded to civilian fields, including logistics, rescue operations, healthcare, and agriculture. UAVs display excellent advantages in addressing the last-mile delivery problem, especially in urban scenarios [6]. Pina-Pardo et al. pointed out that in addressing the traveling salesman problem, the application of UAVs could reduce delivery times by at least 20% [7]. However, due to the limitation of battery performance, the service range of UAVs is relatively limited. Coindreau et al. proposed a truck-and-UAV solution to expand the service range, which reduced delivery times by 34% compared to truck-only deliveries [8]. However, the implementation of UAV logistics as an innovative logistics model has some barriers and constraints. Sah et al. stated that regulations and threats to privacy and security are the key factors that constrain the implementation of delivery via UAVs [9]. Sandbrook raised concerns about the safety of UAV users' data [10]. In addition, the emission problems, noise problems, and environmental issues brought about by UAV-logistics applications have been discussed [11–13].

(2)    Operation and management of UAV logistics

The rapid development of UAV logistics has raised the need for requirements for its management. The current research on the operational management of UAV logistics mainly includes path planning, airport landing and takeoff layout, delivery-mode design, and operational environment planning. For UAV logistics path planning, a collision free path planning algorithm based on the A* algorithm, was proposed by Shi et al. after taking into account the dynamic and random characteristics of the operation environment of UAV logistics. This algorithm achieved an average planning time of 19.45 s when the distribution demand was 5000, and the collision probability rose as the distribution demand continued to rise [14]. Duan et al. identified capacity limitation as the primary issue with the UAV logistics system, combined the memetic algorithm and variable neighborhood descent, proposed a logistics UAV path optimization method, and carried out simulation experiments based on real geographic environments to carry on the logistics UAV path planning for four logistics UAVs to fulfill 200 delivery demands [15]. Zhao et al. suggested a 3D logistics UAV path planning approach based on the APF–RRT* algorithm and contrasted it with the conventional RRT and RRT* algorithms, demonstrating that the proposed APF–RRT* algorithm performed at its best in terms of planning effect and planning time [16]. For logistics UAV landing and takeoff airport layout, based on actual geographic information

for San Francisco, Shavarani et al. integrated the logistics costs associated with UAV takeoff and landing, charging, and UAV purchasing, and presented a logistics UAV takeoff and landing airport and charging station layout concept [17]. German et al. presented a small parcel delivery scheme utilizing eVTOL to deliver from a warehouse in Tracy, California, to the San Francisco Bay Area as a scenario, and presented a logistics UAV takeoff and landing field layout scheme with the goal of maximizing delivery demand [18]. Hong et al. established a mixed integer planning model for logistics UAV charging equipment layout by considering the influence of obstacles on the operation capability of logistics UAVs, and carried out a large-scale simulation experiment in Phoenix City to test the efficacy of the suggested model and methodology [19]. For logistics UAV delivery mode design, Brunner suggested a mode of delivering goods to customers' balconies by logistics UAVs for the last mile of express transportation in urban areas, using GPS to assist logistics UAVs to reach the vicinity of delivery locations, and then using visual navigation equipment to lock the exact delivery location [20]. Li et al. developed a path optimization model for the collaborative delivery of vehicles and UAVs for emergency logistics delivery scenarios and experimentally demonstrated that the collaborative delivery model can effectively improve timeliness and customer satisfaction [21]. For logistics UAV operation environment planning scale operation, Li et al. proposed a logistics UAV route network planning approach based on improved cellular automata and the best-spanning tree algorithm. They also created a route network covering 378 distribution requirements [22]. Salleh et al. established three configurations of urban low-altitude route networks based on a regional scenario in Singapore for UAM development requirements [23].

(3) Airspace capacity assessment

The rapid development of UAV logistics brings challenges to low-altitude airspace management. Considering the expected development tendency of large-scale and routine logistics UAV operations, an urban low-altitude airspace capacity assessment study is critical for ensuring the safe and efficient operation of logistics UAVs. The early airspace capacity assessment studies focused on runways and terminal areas, and mathematical modeling was adopted as the main research method. Bowen and Pearcy proposed a Poisson distribution-based airport runway arrival flow model with reference to ground traffic flow models, which provides a foundation for airport capacity assessment [24]. Janic and Tosic pioneered the establishment of a terminal area capacity assessment model [25]. Computer technology began to develop rapidly in the 20th century, which provided new ideas for airspace capacity assessment research. SIMMOD, RAMS, and TAAM software have been developed to simulate the operation of airports, sectors, and other airspace structures, further improving the reliability and precision of airspace capacity assessment [26,27]. In recent years, data mining and machine learning techniques have become prevalent, and intelligent algorithms have been applied to airspace capacity assessment research [28]. However, the urban low-altitude airspace environment is more complex, and existing airspace capacity assessment methods are difficult to apply directly to low-altitude airspace. Because the aircraft operating in the urban low-altitude airspace environment are mainly vertical takeoff and landing aircraft, such as rotary-wing UAVs, eVTOL, and helicopters, and the operation and management modes are innovative, the well-established capacity assessment models for sector or terminal area operation are not applicable. Meanwhile, UAM is still in the developmental stage, and there is limited operational data from real-world scenarios for research. The current low-altitude airspace capacity assessment is mostly based on the control variable approach for large-scale simulation trials [29–31] to explore the influence of airspace structure [32], route network structure [23], operation interval [33], CNS [34] and other factors on the low-altitude airspace capacity.

To sum up, some outcomes have been achieved with the development of UAM and logistics UAVs. However, there are relatively few studies on low-altitude airspace management for large-scale logistics UAV operations. This paper proposed a capacity assessment method for logistics UAV route networks. The aim is to provide a foundation for the future normalized logistics UAV operation and to ensure the safety and efficiency of

logistics UAVs. This paper presents a research methodology that combines mathematical modeling and simulation verification. The definition of logistics UAV route network operation capacity for the logistics UAV scale operation scenario is proposed, and the logistics UAV operation mode in urban regions is clarified. A methodology for assessing the capacity to operate a logistics UAV route network is developed, taking safety, cost, and noise into account. The route network covering 62 nodes was built to support the simulation experiments. Through multiple groups of comparison experiments, the two key parameters of logistics UAV operation interval and flight speed are analyzed with the evolution rule of the route network operation capacity.

## 3. Methodologies

A logistics UAV operation mode has been identified in this study. The multiobjective optimization model for logistics UAV flight route network capacity assessment is established based on the new operation mode. This model integrates the impact of safety, cost, and noise on logistics UAV operation by establishing the impedance function.

### 3.1. Problem Description

The present route network operation capacity assessment method is oriented to the high frequency, short distance, and multi-cycle urban logistics transportation and distribution needs, considering safety, noise, and cost factors during the logistics UAV operation, aiming to realize the effective assessment of urban logistics UAV delivery network operation capacity. This paper constructs a logistics UAV delivery scenario with reference to the operation mode of the CaiBird post, as shown in Figure 1. CaiBird post is a new type of logistics service platform in China that co-operates with many logistics companies. CaiBird service stations are usually built in neighborhoods, and different logistics companies deliver packages to the stations, where customers can choose to pick up the packages or have them delivered to their homes.

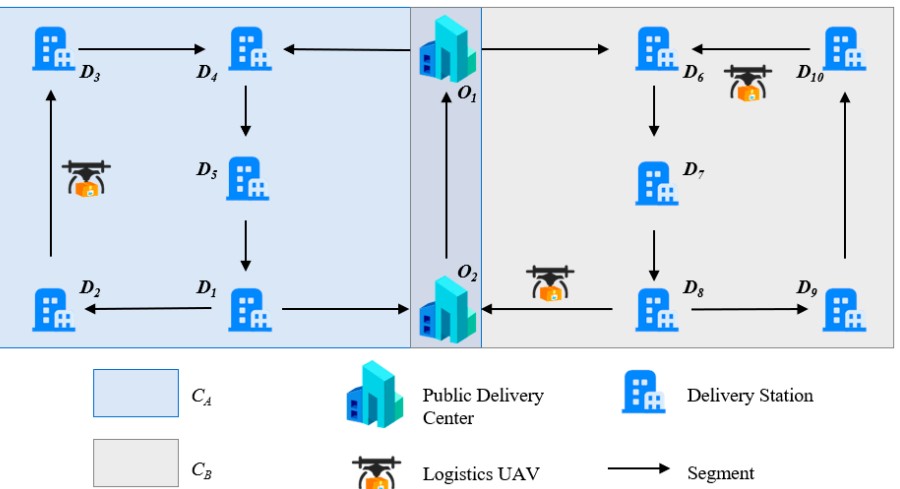

**Figure 1.** Logistics UAV delivery scenario schematic.

The route network is equipped with 2 public delivery centers ($O_1$, $O_2$) and 10 delivery stations $D_i$ ($_i$ = 1, 2, . . ., 7), divided into 2 service communities ($C_A$, $C_B$). Logistics UAVs fly along the route to complete the delivery plan, which consists of multiple directed segments. Each delivery center and delivery station is linked to at least one fly-in segment and one fly-out segment. Considering the demand for transferring and returning goods in the actual logistics transportation, delivery stations and delivery centers are provided with shipping and receiving functions. In order to simplify the calculation, the suggested approach for determining the operational capacity of the route network assumes:

1. Logistics UAVs operate within the specified communities; cross-regional delivery is not allowed, and the public delivery center is at the boundary of each service community and can provide services for multiple community logistics UAVs;
2. All logistics UAVs must operate at a steady speed along a predetermined path in accordance with the delivery plan, keeping a specific safety interval and not allowing any route changes in the middle of the flight. They are all identical in kind and consistent in their performance parameter settings;
3. Logistics UAVs are allowed to land or skip through any public distribution center or delivery station, and the public delivery center or delivery station can only provide service for one logistics UAV at the same time;
4. Logistics UAVs are required to complete takeoff and landing within the specified time, beyond which the delivery plan cannot be executed;
5. Ignoring the effect of weather on logistics UAV operations.

*3.2. Modeling*

3.2.1. Objective Functions

The operational capability of the route network is specified as the maximum logistics UAVs that can be served by all vertiports (including public delivery centers and delivery stations) within the route network in fixed operation time. In addition, it is necessary to consider the impact of safety, cost, and noise factors on logistics UAV operation.

(1) The First Objective

Based on the above operation scenario, the logistics UAVs fly according to the delivery plan. The delivery plan defines the beginning and end vertiports as well as the routes to be flown. Therefore, the number of logistics UAV sorties that can be serviced by the vertiports is defined as the number of delivery plans that can be executed. The first objective of the model is to maximize the number of logistics UAV delivery plans that can be executed in the fixed operation time, as shown in formula (1). In this paper, the fixed operation time is set to 30 s, 60 s, 90 s, and 120 s to analyze the difference in the capacity of the route network at different fixed operation times:

$$MaxN = \sum_{k=1}^{K} f(n_k) \tag{1}$$

where $N$ represents the operation capacity, namely the greatest number of logistics UAV delivery plans that can be executed. $n_k$ represents the delivery plans that are requested to be executed within a fixed operation time. $f(n_k)$ represents the logistics UAV delivery plan as a 0–1 variable, as shown in formula (2):

$$f(n_k) = \begin{cases} 1, \text{delivery plan } k \text{ is executed} \\ 0, \text{delivery plan } k \text{ is not executed} \end{cases} \tag{2}$$

(2) The Second Objective

Referring to the concept of road traffic impedance, consider safety, cost, and noise factors and establish an impedance function for the route network operation. The safety factor is the malfunctioning of the logistics UAVs, which then crash to the ground and cause injuries to people on the ground. The causes of logistics UAV malfunctions include system failures, operational errors, bird strikes, etc. The cost factor refers to the expenses incurred during the logistics UAV operation process, which are mainly related to the flight time and package weight. The longer the flight time and the heavier the package, the higher the cost impedance. The noise factor is the sound produced by the interaction between the rotor rotation and the air during the logistics UAV operation, and the lower the flight altitude,

the stronger the impact of flight noise. The second objective of the model is to minimize the total impedance value in fixed operation time, as shown in formula (3):

$$MinC = \omega_{\text{risk}} \sum_{k=1}^{K} f(n_k) C_{\text{risk}}^k + \omega_{\text{noise}} \sum_{k=1}^{K} f(n_k) C_{\text{noise}}^k + \omega_{\cos t} \sum_{k=1}^{K} f(n_k) C_{\cos t}^k \tag{3}$$

where $C$ represents the total operation impedance value of the route network, which includes the safety impedance $C_{\text{risk}}^k$, noise impedance $C_{noise}^k$, and cost impedance $C_{\cos t}^k$. $\omega_{risk}$, $\omega_{noise}$, $\omega_{\cos t}$ are the weighting coefficients of safety impedance, noise impedance, cost impedance, and $\omega_{\text{risk}} + \omega_{noise} + \omega_{\cos t} = 1$.

In order to realize the effective characterization of the logistics UAV operation environment, the ground environment is equally divided into *rth* grids, as shown in Figure 2:

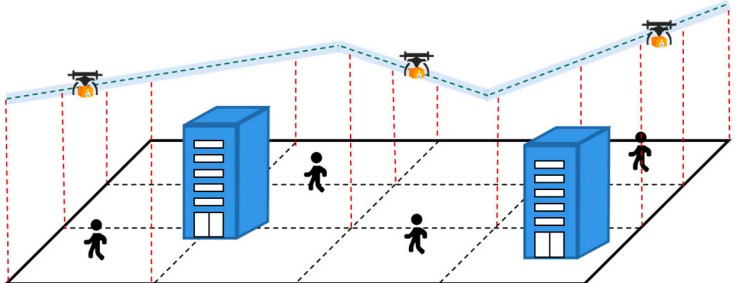

**Figure 2.** Logistics UAV operation environment grid schematic.

(1)  Safety Impedance

The route network safety impedance is shown in formula (4):

$$C_{\text{risk}}^k = P_{crash} N_{people} F_{die} \tag{4}$$

where safety impedance $C_{\text{risk}}^k$ consists of three elements. $P_{crash}$ represents the probability of logistics UAVs crashing in an accident. $N_{people}$ represents the number of logistics UAVs that hit ground people after crashing. $F_{die}$ represents the probability that the ground people will die after being hit by a falling logistics UAV.

① The number of people hit by logistics UAVs.

The number of people hit by logistics UAVs $N_{people}$ is shown in formula (5):

$$N_{\text{people}} = \sum_{r=1}^{n} L_{ij}^r \times \rho_r \times A \tag{5}$$

where $L_{ij}^r$ represents the distance of the segment $L_{ij}$ in the grid $r$. $n$ represents the total number of grids crossed by segment $L_{ij}$. $\rho_r$ represents the ground population density of grid $r$; the calculation is shown in formula (6). A represents the ground area affected by the logistics UAV crash landing, as shown in formula (8):

$$\rho_r = \sum_{b \in B} d_b e^{(1 - r_{rb}^2)} \rho_{\text{avg}} \tag{6}$$

In formula (6), $\rho_{\text{avg}}$ represents the average population density. $r_{rb}$ represents the distance between the center point of building $b$ and the grid $r$. B represents the set of all buildings. $d_b$ represents the delivery demand for building $b$, which is positively correlated with the size of the building; the calculation method is shown in formula (7):

$$d_b = \varepsilon(n_b \times F_b) \tag{7}$$

In formula (7), $\varepsilon$ represents the correlation coefficient. $n_b$ represents the number of grids occupied by building $b$. $F_b$ represents the floor height of building $b$.

$$A = w_{uav} \times \sqrt{l_{uav}^2 + h_{uav}^2} \tag{8}$$

In formula (8), $w_{uav}$ represents the length of the logistics UAV wingspan. $l_{uav}$ represents the length of logistics UAV fuselages. $h_{uav}$ represents the height of logistics UAV fuselages.

② The lethality rate after hitting people

$F_{die}$ represents the probability of ground people dying after being hit by a crashing logistics UAV, as shown in formula (9):

$$F_{die} = \frac{1}{1 + \sqrt{\frac{\alpha}{\mu}\left(\frac{\mu}{E}\right)^{\frac{1}{4s}}}} \tag{9}$$

where $E$ represents the logistics UAV impact kinetic energy, which is calculated as shown in formula (10). $S$ represents the average sheltering factor of the route. $\alpha$, $\mu$ represents fixed parameters in the model.

$$E = \frac{(m_{uav} + m_{delivery}) \times v_{crash}^2}{2} \tag{10}$$

where $m_{uav}$ represents logistics UAV unladen mass. $m_{delivery}$ represents logistics UAV carry package quality. $v_{crash}$ represents logistics UAV crash speed, as shown in formula (11):

$$v_{crash} = [\max(v_{uav}, v_{wind})] + v_{\text{vertical}} \tag{11}$$

where $v_{uav}$ represents logistics UAV flight speed. $v_{wind}$ represents horizontal wind speed. $v_{vertical}$ represents logistics UAV vertical crash speed.

(2)    Noise Impedance

The noise impedance of logistics UAV operations is proportional to population density and distance to logistics UAVs, as shown in formula (12):

$$C_{noise}^k = \sum_{r=1}^{n} \sum_{g \in G} \sum_{f=1}^{F_g} \max[\frac{1}{v_{uav}} L_{ij}^r \rho_g (L_0 - L_{CMFY} - \Delta L_{noise}^{fg}), 0] \tag{12}$$

As shown in Figure 3, $r$ represents the grid with the logistics UAV segment path. $L_{ij}^r$ represents the distance of the segment $L_{ij}$ in the grid $r$. $G$ is the set of gird $g$. $F_g$ represents the floor of the building in grid $g$ if there is no building $F_g = 1$. $f$ represents the floor of the building. $\rho_g$ represents the population density in grid $g$; the calculation method is consistent with $\rho_r$. $L_0$ represents the logistics UAV initial noise value. $L_{CMFY}$ represents the people's acceptable value of logistics UAS noise. $\Delta L_{noise}^{fg}$ represents noise attenuation of the *fth* floor in grid $g$, as shown in formula (13) [35]:

$$\Delta L_{noise}^{fg} = 10\lg(\frac{1}{4\pi d_{fg}^2}) \times (1 + 0.08G_g) \tag{13}$$

where $G_g$ represents the logistics UAV noise barrier factor [36], which indicates the shielding effect of ground buildings and trees from logistics UAV noise. $h_{fg}$ represents the distance between logistics UAV and people affected by noise; the calculation method is shown in formula (14):

$$d_{fg} = \sqrt{(h_{uav} - h_{people})^2 + \Delta x_{rg}^2 + \Delta y_{rg}^2} \tag{14}$$

where $h_{uav}$ represents the vertical distance of logistics UAV from the grid. $h_{people}$ represents the vertical distance of people from the grid. $\Delta x_{rg}$ represents the horizontal distance between grid $r$ and grid $g$. $\Delta y_{rg}$ represents the vertical distance between the grid $r$ and the grid $g$.

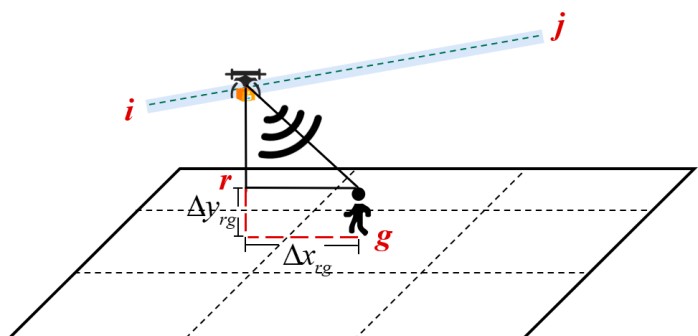

**Figure 3.** Logistics UAV operation noise impact schematic.

(3)    Cost Impedance

The noise impedance of the route network is shown in formula (15):

$$C_{\cos t}^{k} = \sum_{r=1}^{n} \frac{L_{ij}^{r}}{v_{uav}} \times \varepsilon(m_{delivery}) \times p_{power} \tag{15}$$

where, $p_{power}$ represents logistics UAVs' flight cost. $\varepsilon(m_{delivery})$ represents the penalty factor when the package mass carried by the logistics UAV is $m_{delivery}$; the calculation method is shown in formula (16):

$$\varepsilon(m_{delivery}) = \frac{\varepsilon_{\max} - 1}{m_{\max}} \times m_{delivery} + 1 \tag{16}$$

where $\varepsilon_{\max}$ represents the upper limit of $\varepsilon(m_{delivery})$. $m_{\max}$ represents the upper limit of package mass.

3.2.2. Constraint Conditions

(1)    Takeoff

If the delivery plan $k$ takes off from the vertiport $O_i$ is executed, namely $f(n_k) = 1$, logistics UAV must pass through any of the segments that fly away from the vertiport $O_i$. If the delivery plan $k$ from vertiport $O_i$ is not executed, namely $f(n_k) = 0$, logistics UAV must not pass through any of the segments that fly away from vertiport $O_i$, as shown in formula (17):

$$f(n_k) = \sum x_k^{ij}, \ L_{ij} \in L_i^{out} \tag{17}$$

where $x_k^{ij}$ is a 0–1 variable, $x_k^{ij} = 1$ represents delivery plan $k$ passes through the segment $L_{ij}$ and $x_k^{ij} = 0$ represents delivery plan $k$ without passing through the segment $L_{ij}$. $L_i^{out}$ represents the set of segments that fly away from the vertiport $O_i$.

(2)    Landing

If the delivery plan $k$ landing at vertiport $O_j$ is executed, namely $f(n_k) = 1$, logistics UAV must pass through any of the segments that fly into vertiport $O_j$. If the delivery plan $k$ landing at vertiport $O_j$ is not executed, namely $f(n_k) = 0$, logistics UAV must not pass through any of the segments which fly into vertiport $O_j$, as shown in formula (18):

$$f(n_k) = \sum x_k^{ij}, \ L_{ij} \in L_j^{in} \tag{18}$$

where $L_j^{in}$ represents the set of segments which fly into from vertiport $O_j$.

(3)　Fly Across

In the scenario proposed in the paper, takeoff and landing services are available at any node of the route network. If the delivery plan $k$ flies across the vertiport $O_z$, namely not landing when arriving at the vertiport $O_z$, flies across the vertiport $O_z$, then continues to follow the fixed route, the logistics UAV must fly across one vertiport $O_z$ fly into segment and one vertiport $O_z$ fly away segment, as shown in formula (19):

$$\sum x_k^{iz} = \sum x_k^{zj} = \left\{ \begin{array}{l} 1, \text{ fly across} O_z \\ 0, \text{not fly across} O_z \end{array} \right. , \ L_{iz} \in L_z^{in}, L_{zj} \in L_z^{out} \tag{19}$$

where $x_k^{iz}$ is a 0–1 variable, $x_k^{iz} = 1$ represents delivery plan $k$ fly across segment $L_{iz}$ and $x_k^{iz} = 0$ represents delivery plan $k$ not fly across segment $L_{iz}$. $x_k^{zj}$ is a 0–1 variable, $x_k^{zj} = 1$ represents delivery plan $k$ fly across segment $L_{zj}$ and $x_k^{zj} = 0$ represents delivery plan $k$ not fly across segment $L_{zj}$. $L_z^{in}$ is the set of vertiport $O_z$ fly into segment, $L_z^{out}$ is the set of vertiport $O_z$ fly away segment.

(4)　Safety Interval

The distance between the front and rear logistics UAV must be larger than the minimum safety interval. In this paper, the safety interval standard is maintained by setting the logistics UAV takeoff interval, as shown in formula (20):

$$\left| t_k^i - t_{k'}^i \right| \geq T_{airroute}, k \neq k' \tag{20}$$

where $t_k^i$ represents the takeoff time of delivery plan $k$. $t_{k'}^i$ represents the takeoff time of the delivery plan $k\prime$. $T_{airroute}$ represents the takeoff interval time between the two logistics UAVs, before and after.

(5)　Fly Across Time

The time of logistics UAV flying across the vertiport is calculated as formula (21):

$$t_k^{j'} = t_k^i + \triangle t_k^{ij} \tag{21}$$

where $t_k^{j'}$ represents the time that logistics UAV flying across the vertiport $O_j$. $\triangle t_k^{ij}$ represents the time required for a logistics UAV to fly over segment $L_{ij}$.

(6)　Load Limitation

The mass of logistics UAVs cannot exceed the upper limit of the takeoff mass after loading the package, as shown in formula (22):

$$m_{uav} + m_{delivery} \leq m_{\max} \tag{22}$$

where $m_{uav}$ represents the mass of logistics UAV without packages. $m_{delivery}$ is the mass of the package. $m_{\max}$ represents the upper limit of logistics UAV takeoff mass.

(7)　Flight Speed Limitation

Logistics UAV flight speed must be between the lower and upper flight speed limits, as shown in formula (23):

$$v_{\min} \leq v_{uav} \leq \max(v_{wind}, v_{\max}) \tag{23}$$

where $v_{uav}$ represents the logistics UAV flight speed. $v_{\min}$ represents the lower limit of logistics UAV flight speed. $v_{\max}$ represents the upper limit of logistics UAV flight speed. $v_{wind}$ represents wind speed.

(8)  Flight Distance Limitation

The total flight distance of logistics UAVs executing each delivery plan must be less than the flight distance limit, as shown in formula (24):

$$\sum L_{ij} \leq L_{ij}^{\max} \tag{24}$$

where $L_{ij}^{\max}$ represents the upper limit of logistics UAV flight distance.

The complete model is shown in formula (25):

$$
\begin{aligned}
&MaxN = \sum_{k=1}^{K} f(n_k) \\
&MinC = \omega_{\text{risk}} \sum_{k=1}^{K} f(n_k) C_{\text{risk}}^k + \omega_{\text{noise}} \sum_{k=1}^{K} f(n_k) C_{\text{noise}}^k + \omega_{\cos t} \sum_{k=1}^{K} f(n_k) C_{\cos t}^k \\
&s.t. \begin{cases}
C_{risk}^{ij} = P_{uav} N_{people} F_{die} \\
C_{noise}^k = \sum_{r=1}^{n} \sum_{g \in G} \sum_{f=1}^{F_g} \max\left[\frac{1}{v_{uav}} L_{ij}^r \rho_g (L_0 - L_{CMFY} - \Delta L_{noise}^{fg}), 0\right] \\
C_{\cos t}^k = \sum_{r=1}^{n} \frac{L_{ij}^r}{v_{uav}} \times \varepsilon(m_{delivery}) \times p_{power} \\
f(n_k) = \sum x_k^{ij}, \ L_{ij} \in L_i^{out} \\
f(n_k) = \sum x_k^{ij}, \ L_{ij} \in L_j^{in} \\
\sum x_k^{iz} = \sum x_k^{zj} = \begin{cases} 1, \ \text{fly across} O_z \\ 0, \ \text{not fly across} O_z \end{cases}, \ L_{iz} \in L_z^{in}, L_{zj} \in L_z^{out} \\
|t_k^i - t_{k'}^i| \geq T_{airroute}, k \neq k' \\
t_k^{j'} = t_k^i + \triangle t_k^{ij} \\
m_{uav} + m_{delivery} \leq m_{\max} \\
v_{\min} \leq v_{uav} \leq \max(v_{wind}, v_{\max}) \\
\sum L_{ij} \leq L_{ij}^{\max}
\end{cases}
\end{aligned}
\tag{25}
$$

where $f(n_k)$, $x_k^{ij}$, $x_k^{ij}$, $x_k^{iz}$, and $x_k^{zj}$ are binary variables; $C_{\text{risk}}^k$, $C_{\text{noise}}^k$, and $C_{\cos t}^k$ are quantitative variables; and $t_k^i$, $t_{k'}^i$, $m_{uav}$, $m_{delivery}$, and $v_{uav}$ are quantitative variables.

## 4. Algorithm

The Non-dominated Sorting Genetic Algorithm-II (NGSA-II) algorithm is utilized to solve the proposed logistics UAV route network capacity assessment multiobjective model. The principles and execution of the NGSA-II algorithm are presented. NGSA-II is a typical heuristic algorithm with the advantages of fast convergence and high efficiency, which is widely used in solving multiobjective optimization problems. Compared with the NGSA algorithm, the NGSA-II algorithm has lower complexity. NSGA-II reduces the algorithm complexity from $O(mN^3)$ to $O(mN^2)$ by a non-dominated sorting method, where $m$ is the number of objective functions and $N$ is the population size. Meanwhile, NGSA-II uses crowding and crowding comparison operators instead of the fitness-sharing strategy that requires the specification of a sharing radius, which helps to maintain the diversity of the population [37]. Figure 4 illustrates the algorithm flow.

Non-dominance sorting is a stratification of all individuals in the population by the number of individuals that can be dominated. As shown in Figure 4, parent populations generate offspring populations through selection, crossover, and mutation. Then, all populations are to be stratified by a non-dominated sorting method. If the set of individuals cannot be dominated by other individuals, they are sorted into Rank 1, the Pareto frontier. If the set of individuals can be dominated by one individual, they are sorted into Rank 2, and so on. If 50% of the individuals (above the red line) are retained as the new parent population, Rank 1 and Rank 2 will be retained directly. Since some individuals in Rank 3 are outside the 50% range, Rank 3 is further reordered by calculating the crowding distance. Crowding distance is a standard for determining the degree of aggregation near individuals

of the same rank. Large crowding distances indicate dense population distribution, which makes the algorithm readily reach local optimal solutions and allows for direct rejection.

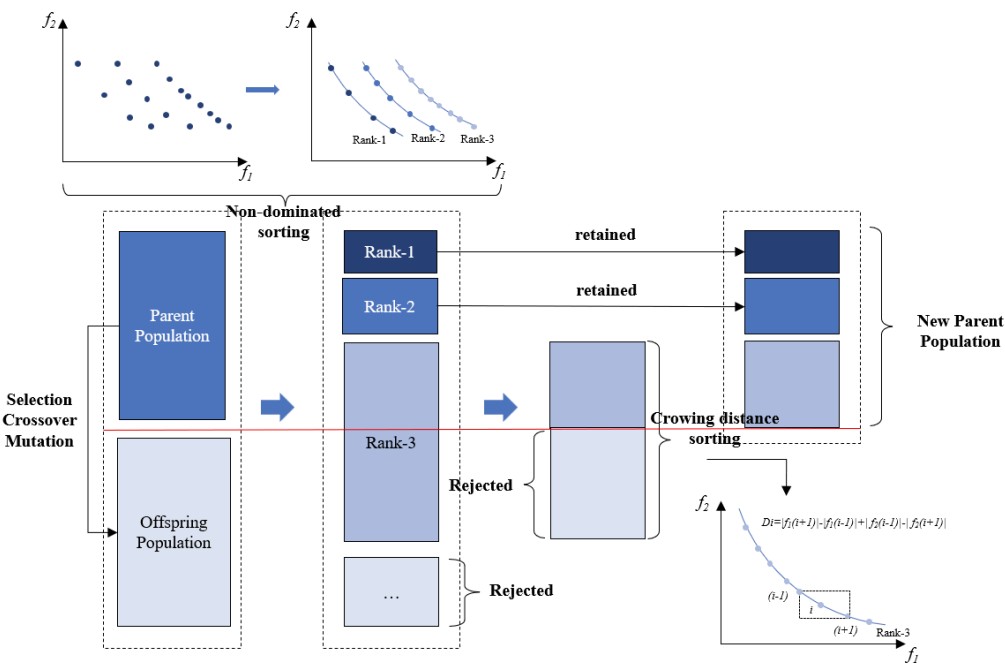

**Figure 4.** NGSA−II algorithm schematic diagram [37].

*NGSA-II Algorithm Genetic Mechanism Design*

(1)　Coding

　　Generating *K* logistics UAV delivery plan serial numbers and randomizing them, as shown in Figure 5. Meanwhile, the route and package weight corresponding to each delivery plan is generated, as shown in Table 1.

| $n_1$ | $n_{k-1}$ | $n_3$ | $n_2$ | $n_4$ | $n_k$ | $\cdots$ | $n_5$ |
|---|---|---|---|---|---|---|---|
| 1 | k−1 | 3 | 2 | 4 | k | ... | 5 |

**Figure 5.** Initial individual coding.

**Table 1.** Delivery plan information.

| Delivery Plan Serial Number | Route | Package Weight (kg) |
|---|---|---|
| 1 | $O_1 \rightarrow O_2$ | 2 |
| $k-1$ | $O_2 \rightarrow O_3 \rightarrow O_4$ | 1 |
| 3 | $O_2 \rightarrow O_3 \rightarrow O_j$ | 2.5 |
| 2 | $O_1 \rightarrow O_3 \rightarrow O_4 \rightarrow O_j$ | 0.5 |
| 4 | $O_3 \rightarrow O_4$ | 3 |
| $k$ | $O_4 \rightarrow O_5 \rightarrow O_j$ | 1.5 |
| $\cdots$ | $\cdots$ | $\cdots$ |
| 5 | $O_1 \rightarrow O_2 \rightarrow O_3$ | 3 |

(2)　Genetic Operations

　　The genetic operation in the NGSA-II algorithm is similar to the traditional Genetic Algorithm (GA) in that it takes three approaches to generate offspring populations, namely

selection, crossover, and mutation. The roulette mechanism is adopted in this paper for the selection operation. When the delivery plan serial number is in crossover and mutation operation, the corresponding route and the package weight carried change in order with the serial number.

The crossover probability $P_c$ and mutation probability $P_m$ are the keys to determining the speed of convergence and stable performance of NSGA-II algorithms. This paper applies the adaptive adjustment mechanism of genetic operators to reduce the probability of good operators being selected for crossover and mutation and to increase the probability of poor operators being selected for crossover and mutation to ensure the gradual and stable improvement of the overall population level. The crossover probability $P_c$ is calculated as formula (26). The mutation probability $P_m$ is calculated as formula (27) [38]:

$$p_c = \begin{cases} 0.1 \times \sqrt{\frac{(1-N'_{rank}) \times G}{(R-1) \times N}} + 0.9 & , N'_{rank} < N_{avg} \\ 1 & , N'_{rank} \geq N_{avg} \end{cases} \tag{26}$$

where $N'_{rank}$ represents the value of the paired crossover operator with the smaller non-dominated rank. $N_{avg}$ represents the average value of the current population's individual non-dominance rank. $G$ represents the current population iteration number. $R$ represents the maximum value of the current population's individual non-dominance rank. $N$ represents the maximum number of iterations.

$$p_m = \begin{cases} 0.1 \times \sqrt{\frac{-(N_{rank}-1) \times G}{(R-1) \times N}} & , N_{rank} < N_{avg} \\ 0.1 & , N_{rank} \geq N_{avg} \end{cases} \tag{27}$$

where $N_{rank}$ represents the non-dominated rank of the mutation operator.

The NGSA-II algorithm's particular stages are as follows [Algorithm 1]:

---

**Algorithm 1.** NGSA-II Algorithm solving process [37]

---

**Input**: The maximum amount of iterations $N_{max}z$
**Output**: Final population Pareto solution set $\mathbb{P}_{N_{max}}$
$N = 0$, $\mathbb{P}_N = \{\mathbf{x}_1, \mathbf{x}_2, \mathbf{x}_3, \cdots, \mathbf{x}_n\}$     // Generating an initial population with population size $\boldsymbol{n}$
**While** $N \leq N_{max}$
    $C_1 = \text{calCost1}(\mathbb{P}_N)$               // Calculating the objective function
    $C_2 = \text{calCost2}(\mathbb{P}_N)$
    $\mathbb{P}_N = \text{nonDominatedSorting}(\mathbb{P}_N, C_1, C_2)$       // Non-dominance sorting
    $\mathbb{P}_N = \text{crowdingDisSorting}(\mathbb{P}_N)$               // Sorting by crowding distance
    $\mathbb{P}_{N+1} = \mathbb{P}_N[: z]$               // Retaining the top $z$ individuals in $\mathbb{P}_N$
    **While** $\text{len}(\mathbb{P}_{N+1}) < n$
        $x_a, x_b = \text{rouletteChoice}(\mathbb{P}_N)$       // Select two individuals from the parent population
        $P_c = \text{calCrossProb}(x_a, x_b)$       // Calculating adaptive crossover probabilities
        $P_m = \text{calMutateProb}(x_a, x_b)$       // Calculating adaptive mutation probabilities
        $x'_a, x'_b = \text{geneticOperator}(x_a, x_b, P_c, P_m)$
        $\mathbb{P}_{N+1}.\text{append}(x'_a, x'_b)$
    $N = N + 1$
**End While**
**Return** $\mathbb{P}_{N_{max}}$

---

## 5. Analysis

A logistics UAV route network based on real geographic information data is established. Comparative experiments are performed to analyze the result of logistics UAV route network capacity. Particular focus is placed on the impact of two key parameters, flight speed and safety interval, on the route network capacity.

### 5.1. Parameters Setting

(1)  Simulation Environment

To verify the effectiveness of the logistics UAV route network operation capacity assessment model and algorithm, Python 3.9.10 was utilized for simulation experiments. A region in Nanjing, China, is selected as the experimental scenario, as shown in Figure 6a, to construct a logistics UAV route network, as shown in Figure 6b. The route network has 62 vertiports, including 53 end delivery stations and 9 public delivery stations, divided into 4 communities. Logistics UAVs fly on fixed headings without two-way segments.

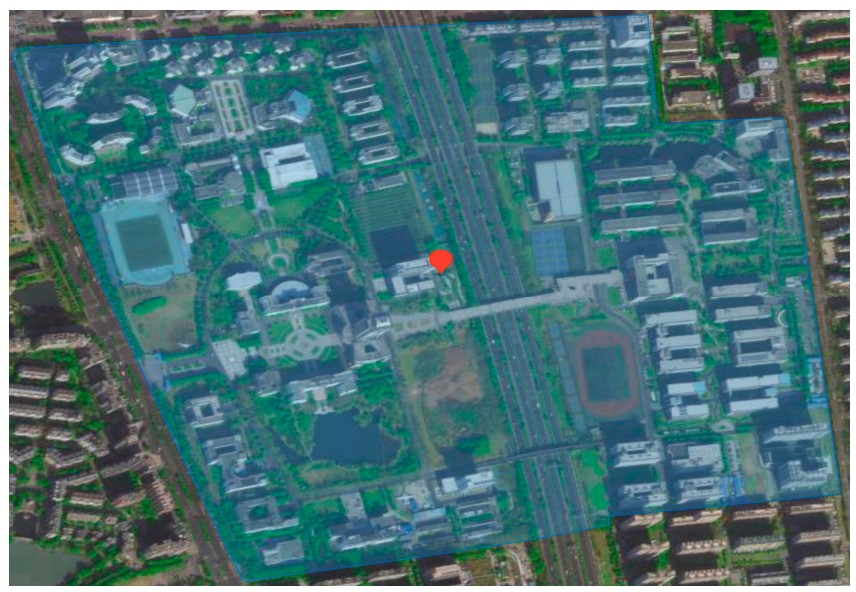

(**a**)

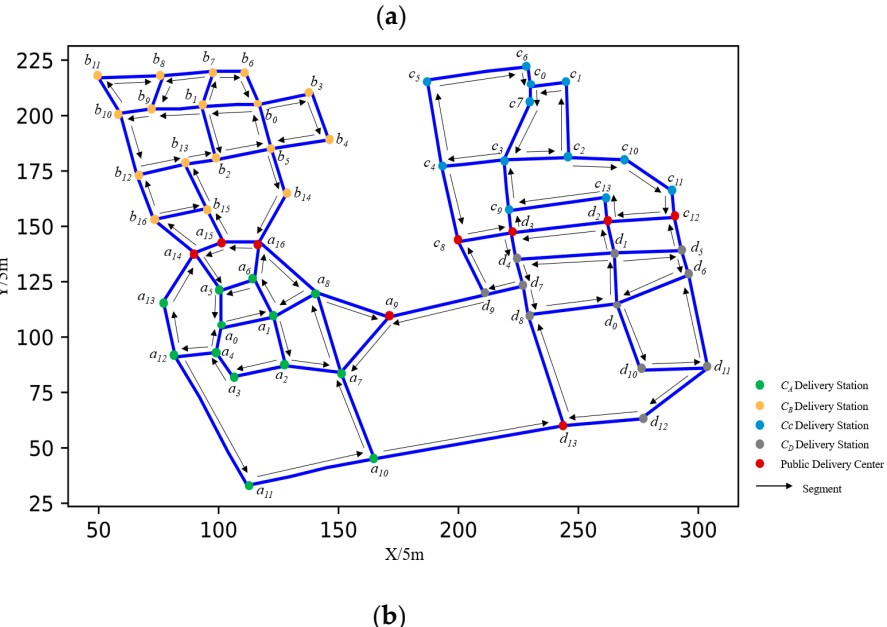

(**b**)

**Figure 6.** (**a**) A region in Nanjing, China. (**b**) Logistics UAV route network in an area of Nanjing, China.

Logistics UAV operation noise impedance is directly related to the height of the route network; refer to the literature [23]. The route network constructed is planned at 50 m above the ground road, and the noise impedance value calculation results in the region shown in Figure 7; the darker the color, the higher the noise impedance value. According to Figure 7, the noise impedance value and building height are directly inversely correlated; the higher

the building height, the closer it is to the logistics UAV path, the more pronounced the noise interference, and the higher the noise impedance value. The distance between the outdoor open area and the logistics UAV route is larger, and the noise impedance value is relatively small.

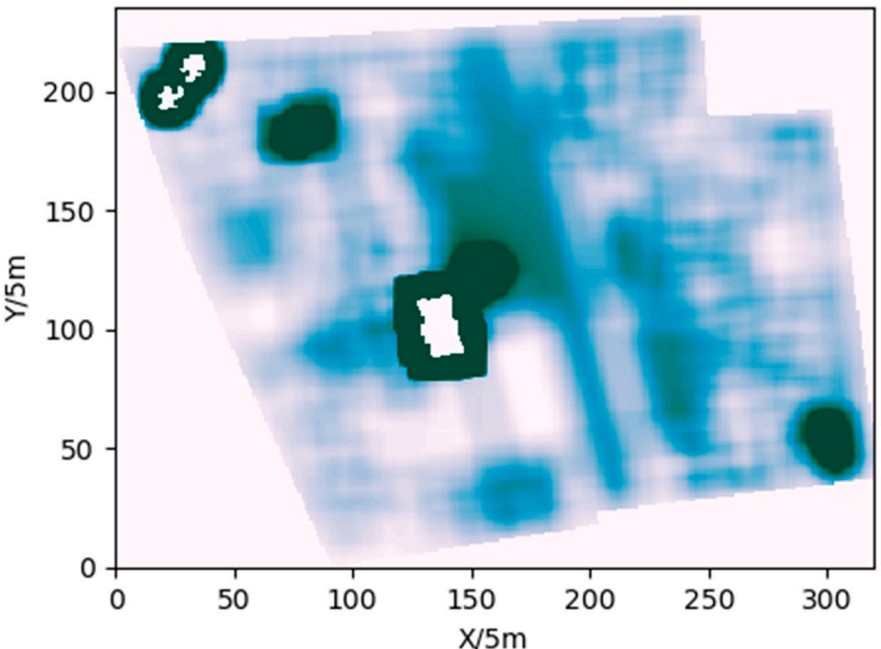

**Figure 7.** Noise impedance distribution in an area of Nanjing, China.

Population density is related to building distribution. People inside the buildings are clustered, and outdoors are relatively loose; the population density in the region is shown in Figure 8; the darker the color indicates that the population density is greater. According to Figure 8, residential neighborhoods and office building areas are densely populated, parks and football fields are relatively loosely populated, and lake areas are almost devoid of people.

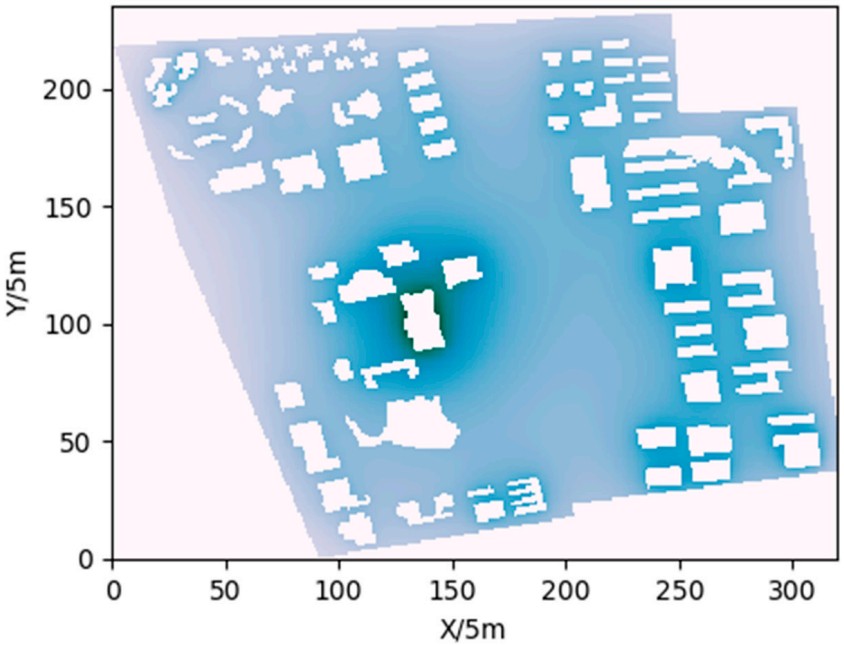

**Figure 8.** Population density distribution in an area of Nanjing, China.

(2)    Parameters

For the logistics UAV delivery scenario, the model parameters are set according to the previous related research [39] and the performance of some multicopter logistics UAVs, such as Meituan FP400, EHang216, and Antwork TR7S, as shown in Table 2.

**Table 2.** Parameter setting [39].

| Parameter | Meaning | Value |
|---|---|---|
| $w_{uav}$ | The length of logistics UAV wingspan | 1 m |
| $l_{uav}$ | The length of logistics UAV fuselages | 0.5 m |
| $h_{uav}$ | The height of logistics UAV fuselages | 0.5 m |
| $m_{uav}$ | Logistics UAV unladen mass | 4 kg |
| $m_{delivery}$ | The mass of logistics UAV carried package | [0, 3] kg |
| $m_{max}$ | The upper limit of logistics UAV takeoff mass | 7 kg |
| $v_{uav}$ | Logistics UAV flight speed | 10 m/s |
| $T_{airroute}$ | The takeoff interval time between the two logistics UAVs before and after | 1.5 s |
| $\varepsilon_{max}$ | The upper limit of $\varepsilon(m_{delivery})$ | 3 |
| $S$ | The average sheltering factor of the route | 0.5 |
| $\alpha$ | Fixed parameter | $10^6$ J |
| $\mu$ | Fixed parameter | 100 J |
| $L_{CMFY}$ | The people's acceptable value of logistics UAS noise | 30 Db |
| $L_{ij}^{max}$ | The upper limit of logistics UAV flight distance | 4 km |
| $\omega_{risk}$ | Safety impedance weighting coefficients | 0.4 |
| $\omega_{noise}$ | Noise impedance weighting coefficients | 0.3 |
| $\omega_{cost}$ | Cost impedance weighting coefficients | 0.3 |
| $Z$ | The number of retained individuals | 100 |
| $N_{max}$ | The maximum number of iterations | 1000 |

*5.2. Results Analysis*

5.2.1. Operation Capacity Analysis

The operation capacity of a logistics UAV route network with operation times of 30 s, 60 s, 90 s, and 120 s is solved using the above model and parameter settings, and the experiment is repeated 20 times for each operation time, and the optimal solution is selected to plot the Pareto front, as shown in Figure 9. The operation capacity and the total impedance value show a linear relationship with similar angular coefficients, and as the total impedance value gradually rises, the operation capacity reaches the inflection point, indicating that the route network operation capacity can continue to optimize the space for improvement gradually decreases. Meanwhile, the Pareto fronts at different operation times are uniformly distributed, indicating that the NGSA-II algorithm maintains the diversity of the population during the solution process.

The optimal solution set among 20 replicate experiments is selected to draw an iterative graph of the operation capacity, as shown in Figure 10. The population optimal solution, namely the optimal operation capacity, and the population average solution, namely the average operation capacity, are outputs for each iteration to comprehensively assess the operation capacity of the logistics UAV route network. The optimal operation capacity of the route network in 30 s, 60 s, 90 s, and 120 s is 414, 523, 599, and 711, respectively, while the average operation capacity is 392, 497, 580, and 688, respectively, and the optimal operation capacity is higher than the average operation capacity by about 20 sorties. When

the operation time is 30 s, the optimal solution could be obtained with 400–500 iterations, and when the operation time increases to 90 s and 120 s, the algorithm begins to converge with more than 500 iterations, indicating that with the increase of operation time, the complexity of computing increases, and the convergence speed of algorithm slows down and the optimization magnitude gradually decreases, reflecting the reasonableness of the proposed logistics UAV route network operation capacity assessment model.

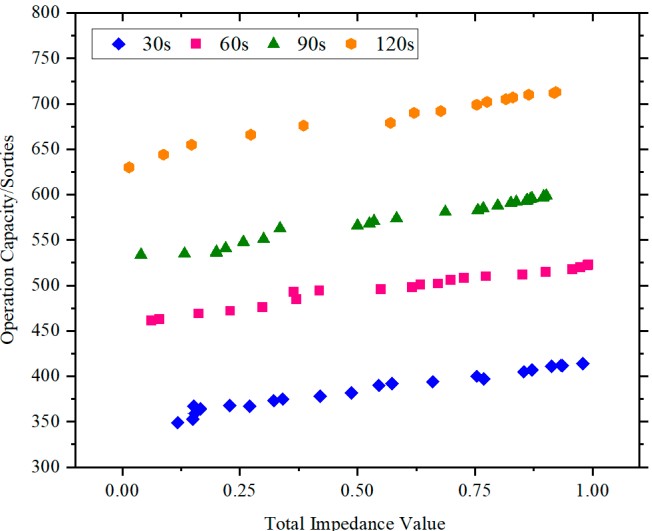

**Figure 9.** The Pareto front.

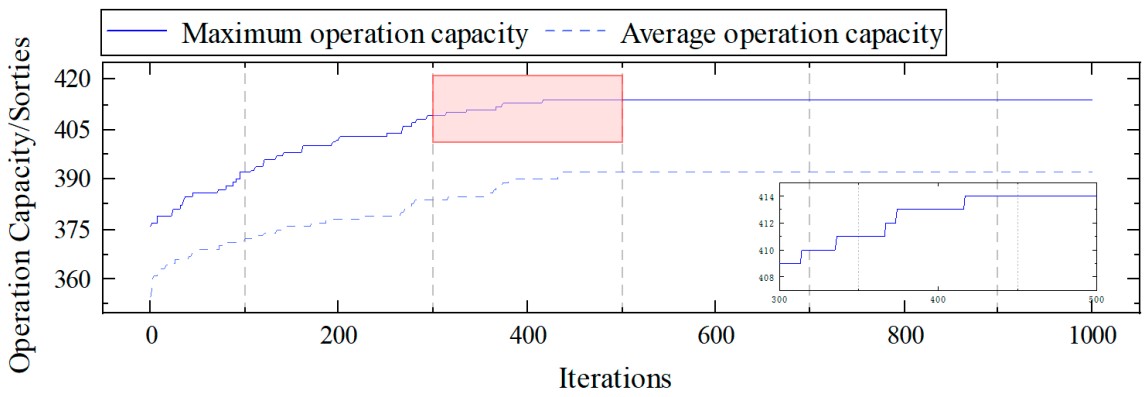

(**a**) Operation time: 30 s

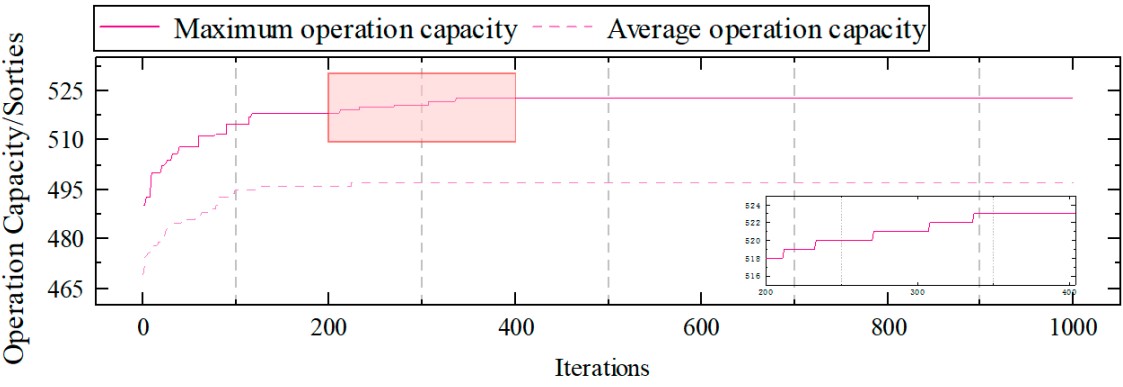

(**b**) Operation time: 60 s

**Figure 10.** *Cont.*

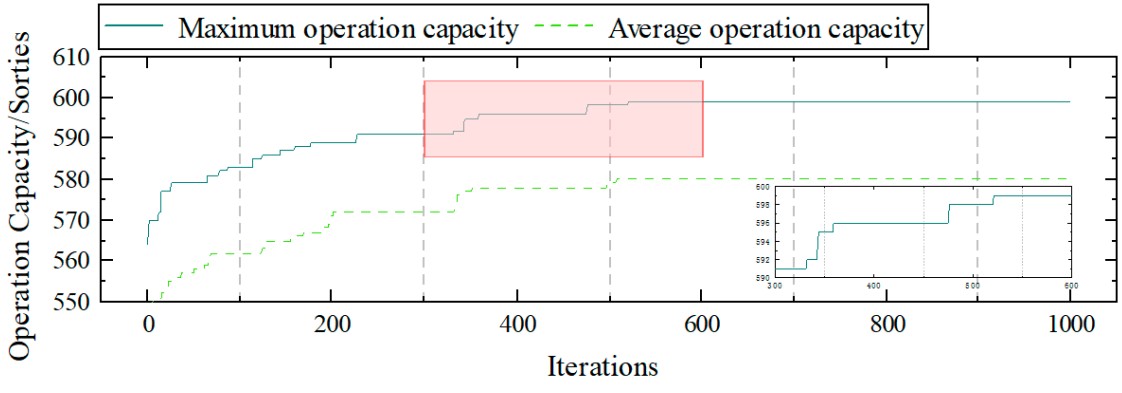

(**c**) Operation time: 90 s

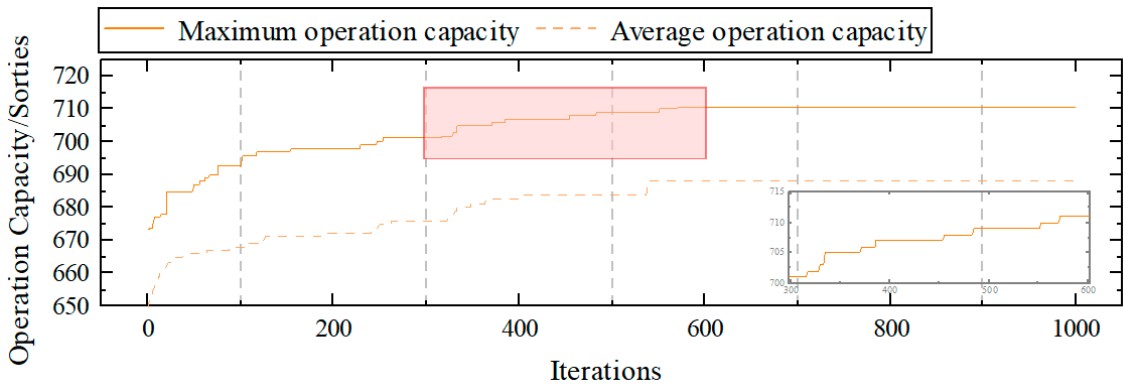

(**d**) Operation time: 120 s

**Figure 10.** Logistics UAV route network operation capacity iteration for different operation times.

5.2.2. Key Parameter Analysis

(1) Safety Interval

The safety interval mentioned in this paper relates to the minimum distance that the two logistics drones in front and behind must maintain when operating in the route network [40]. Maintaining safety intervals is the basic requirement to ensure the logistics UAV operates safely. The route network operation capacity at different operation times with safety intervals of 15 m, 20 m, 25 m, and 30 m is assessed by considering the performance of multicopter logistics UAV, and each group of experiments is repeated 20 times, and the average value is obtained to plot average operation capacity at different safety intervals with the average total impedance value change, as shown in Figure 11.

According to Figure 11, When the operation duration is fixed, the total impedance value drops as the safety interval grows and the route network's operation capacity steadily declines. Moreover, when the safety interval is increased from 25 m to 30 m, the operation capacity decreases significantly less than when the safety interval is increased from 15 m to 20 m and from 20 m to 25 m. It is shown that in the scenario proposed in this paper, with the increase of safety interval, the sensitivity of logistics UAV route network operation capacity to safety interval gradually decreases; namely, when the safety interval reaches a certain level, the influence of safety interval on route network operation capacity gradually decreases.

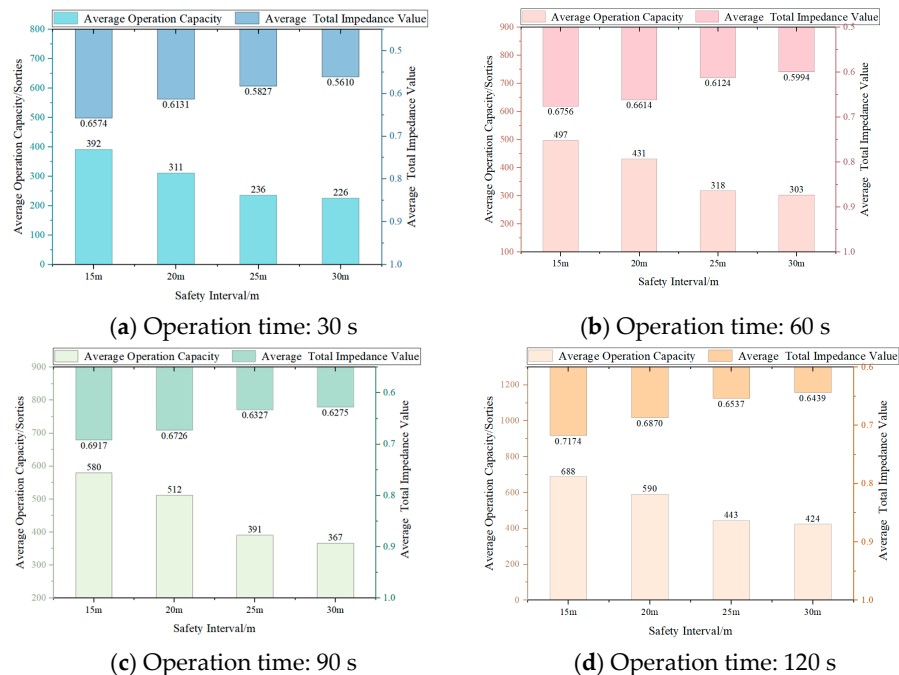

(**a**) Operation time: 30 s

(**b**) Operation time: 60 s

(**c**) Operation time: 90 s

(**d**) Operation time: 120 s

**Figure 11.** Relationship between safety intervals and operation capacity at different operation times.

To analyze the assessment of the operation capacity of the logistics UAV route network in relation to the safety interval, the 'operation time-safety interval-average operation capacity' and 'operation time-safety interval-average total impedance value' graphs were plotted based on the above experimental data, as shown in Figures 12 and 13. A comprehensive analysis of the two key parameters of operation time and safety interval shows that as the operation time increases and the safety interval decreases, the average operation capacity tends to increase. When the operation time is smaller while the safety interval is larger, the average operation capacity is affected more by the operation time; when the operation time is larger while the safety interval is smaller, the average operation capacity is affected more by the safety interval. The smaller variation in the average total impedance value compared to the average operation capacity indicates that the impedance value required to achieve the same increase in operation capacity is decreasing under the influence of marginal effects.

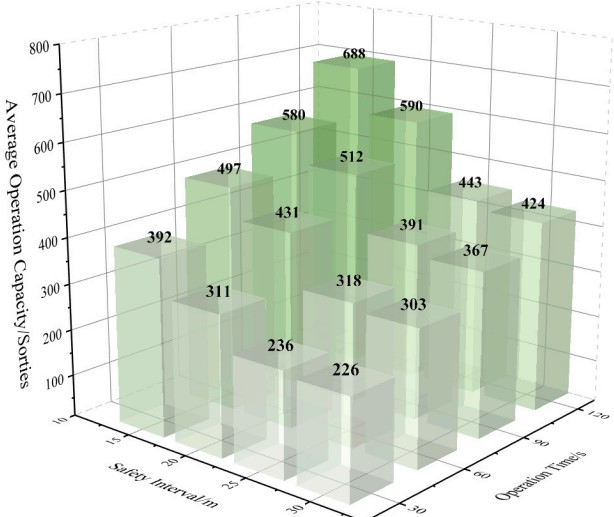

**Figure 12.** Relationship between operation time, safety interval, and average operation capacity.

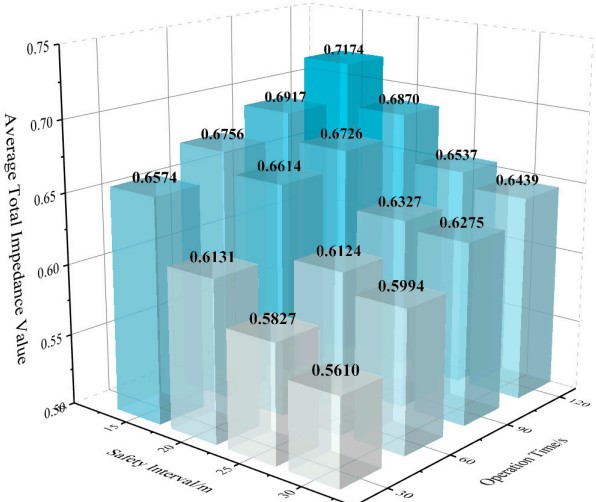

**Figure 13.** Relationship between operation time, safety interval, and average total impedance value.

(2) Flight Speed

According to the scenario developed in this study, logistics UAVs fly at a constant speed throughout the route network, and with reference to the performance parameters of Meituan FP400 logistics UAV, the research analyses the trend of the operation capacity of the route network when the safety interval is fixed at 20 m, and the flight speed of logistics UAVs is increased from 10 m/s to 15 m/s in 0.5 m/s increments. Each group of experiments was repeated 20 times, the average operation capacity and the maximum operation capacity were recorded, and the average of the average operation capacity was obtained to plot the change in flight speed and the average operation capacity. As shown in Figure 14, the average operation capacity rises gradually as the flight speed increases; as the operating time rises, the average operation capacity at the same flying speed improves at a slower rate.

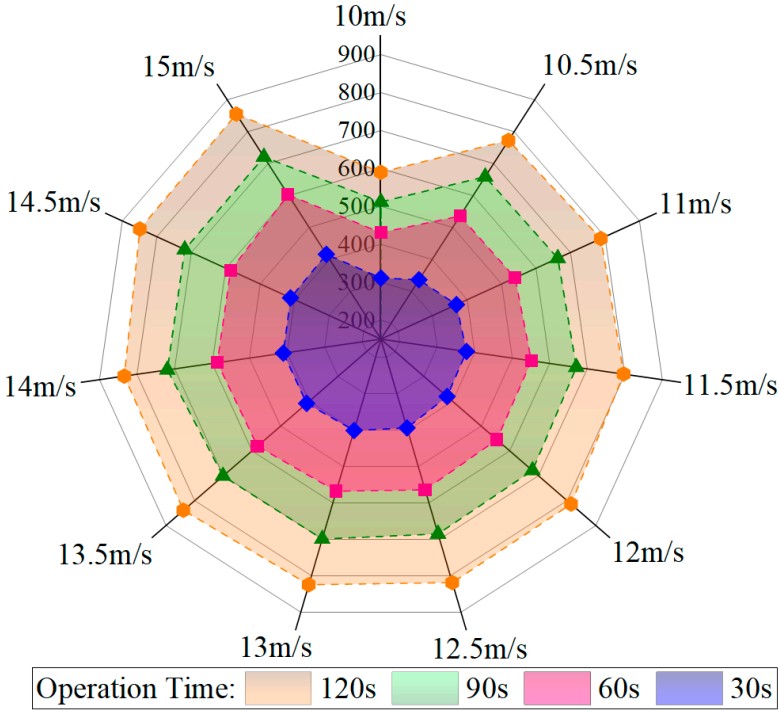

**Figure 14.** Relationship between flight speed and logistics UAV route network operation capacity.

To explore the effect of logistics UAV flying speed on route network operation capacity, based on the above experimental data, the average value of maximum operation capacity and average operation capacity of each group of experiments are taken to plot the relationship between flight speed and operation capacity at different operation time, as shown in Figure 15. According to Figure 15, when the logistics UAV flight speed is increased from 10 m/s to 10.5 m/s, the operation capacity rises at the maximum rate, and as the flight speed increases, the overall rate of rise gradually slows down. Moreover, as the operation time increases, the rate of increase in operation capacity grows. It indicates that when logistics UAVs fly at speeds between 10 m/s and 10.5 m/s, the operation capacity of the route network is influenced by the flight speed, and its influence gradually strengthens as the operation time rises, while the maximum operation capacity is greater than the average operation capacity by about 20 sorties.

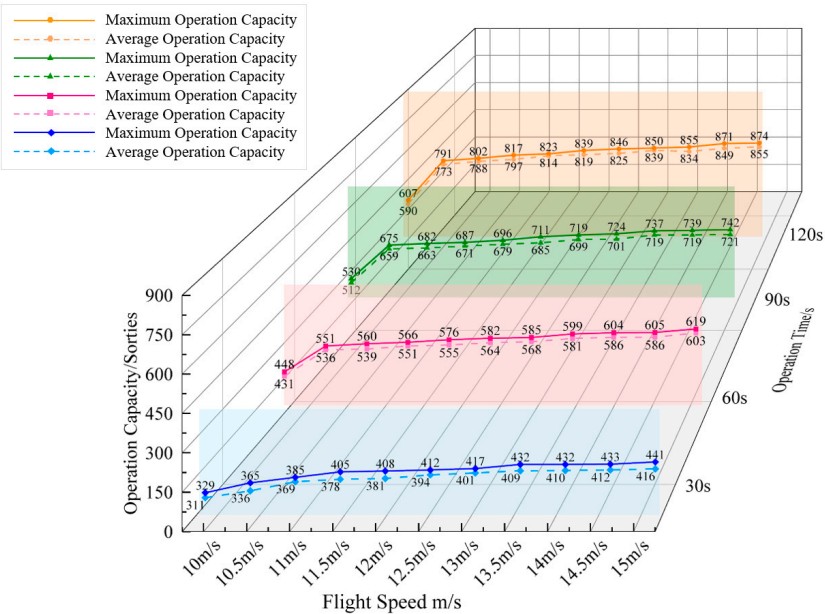

**Figure 15.** Relationship between flight speed, the average maximum operation capacity, and average operation capacity.

In the Analysis section, two main parameters, flight speed and safety interval, are selected for multiple comparative experiments. Reference [41] also establishes a capacity assessment model for logistics UAV routes and selects the safety interval as the key parameter to be analyzed. In reference [41], the capacity of the route network decreases with increasing safety intervals, and the rate of decrease slows down, which is similar to the findings of this paper, which verified the reliability. Moreover, reference [41] does not concern the effect of flight speed on the route network's capacity, and this study is more comprehensive than reference [41].

## 6. Conclusions

This paper proposes an urban low-altitude logistics UAV route network operation capacity assessment method in order to respond to the development trend of logistics UAVs; the following are the contributions:

1.  This paper clarifies the operation mode of logistics UAVs in urban low-altitude airspace and defines the operation capacity of logistics UAV route networks as the maximum sortie of logistics UAVs that can be served during the operation time of all vertiports in the route network. A bi-objective optimization model for assessing the logistics UAV route network operation capacity is established, considering safety, cost, and noise factors. The first objective is to maximize the logistics UAV delivery

plan that can be executed during operation time. The second objective is to minimize the total impedance value;

2.  The route network is established utilizing real-world geographic data with a total of 62 vertiports, including 53 end delivery stations and 9 public delivery stations, divided into 4 communities. Based on the above model and experimental scenarios, the NSGA-II algorithm is adopted to solve the model with operation times of 30 s, 60 s, 90 s, and 120 s, respectively. As the operation time increases, the optimal route network capacity increases from 414 to 711, and the algorithm convergence rate slows down, indicating the reasonableness of the proposed model and algorithm;

3.  Comparative experiments were designed for the key parameter of the safety interval to assess the logistics UAV route network operation capacity at different operating times with safety intervals of 15 m, 20 m, 25 m, and 30 m. Experiments reveal that when the safety interval rises, the average operation capacity of the route network rapidly declines, and the sensitivity to the safety interval decreases accordingly. In addition, the average total impedance value varies less than the average operation capacity, indicating that the impedance value required to achieve the same increase in operation capacity is decreasing under the influence of marginal effects;

4.  Multiple group experiments are carried out to analyze the trend of the route network operation capacity when the flight speed of logistics UAVs is increased from 10 m/s, in 0.5 m/s increments, to 15 m/s, with the goal to expand the relationship between the flight speed of logistics UAVs and the route network operation capacity. The experiments show that as the flight speed increases, the average operation capacity gradually rises, especially when the logistics UAV flight speed is between 10 m/s and 10.5 m/s. The route network operation capacity is influenced by the flight speed, and its influence gradually strengthens as the operation time rises.

This paper proposed a logistics UAV route network operation capacity assessment method. However, the urban environment is complicated, and the logistics UAV operation scenario has meteorological and other dynamic influencing factors interference. In the future, we will try to further study the logistics UAV route network dynamic capacity assessment method through a combination of computer simulation and mathematical modeling methods. Attempts to simulate random influences in the urban low-altitude environment, such as wind shear and bird interference, are made through computer simulation techniques. In addition, we are trying to establish a dynamic network of logistic UAV route networks, which can be based on real-time changes in distribution demand, and studying the dynamic route network capacity assessment methodology. Overall, the study on the capacity assessment of the logistics UAV route network will play a significant role in the planning and scheduling of logistics demand in the real world, especially for the future urban air-ground coordinated logistics and distribution mode. Scientific assessment of the capacity of low-altitude airspace is an important prerequisite for the efficient allocation of air and ground transport resources. Moreover, this study will also provide support for urban low-altitude airspace management and provide a reference value for the development of a reasonable logistics UAV management plan.

**Author Contributions:** Conceptualization, H.Z.; methodology, J.Y. and C.N.; software, F.W.; validation, J.Y., H.L. and G.Z.; formal analysis, J.Y. and F.W.; investigation, J.Y. and C.N.; resources, J.Y., H.Z. and H.L.; writing—original draft preparation, J.Y.; writing—review and editing, J.Y. and F.W.; visualization, J.Y. and F.W. All authors have read and agreed to the published version of the manuscript.

**Funding:** This research was funded by the National Natural Science Foundation of China (71971114).

**Data Availability Statement:** Data will be made available on request.

**Conflicts of Interest:** The authors declare no conflict of interest.

## Abbreviations

| Abbreviation | | Meaning |
|---|---|---|
| UAM | Urban Air Mobility | A new three-dimensional transport system with integrated manned/unmanned aerial vehicle operations on an urban or intercity scale [1,2]. |
| Logistics UAV | Logistics Unmanned Aerial Vehicle | Unmanned Aerial Vehicles (UAVs) used in logistics and transport applications, usually carrying parcels [15]. |
| eVTOL | Electric Vertical Takeoff and Landing | An innovative aircraft, vertical takeoff and landing aircraft with an electric power engine [18]. |
| Logistics UAV route | Logistics Unmanned Aerial Vehicle route | An airway serving logistics UAVs which usually requires pre-planning [36]. |
| NSGA-II | Non-dominated Sorting Genetic Algorithm-II | One of the most popular multiobjective genetic algorithms, which reduces the complexity of non-inferiority sorting genetic algorithms [37]. |
| GA | Genetic Algorithm | A stochastic search algorithm that draws on natural selection and natural genetic mechanisms in biology [37]. |
| - | Safety Interval | Minimum distance between front and rear logistics UAVs [40]. |

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
