# Peer review of "An Operational Capacity Assessment Method for an Urban Low-Altitude Unmanned Aerial Vehicle Logistics Route Network"

_drones, doi:10.3390/drones7090582_

Round 1
Reviewer 1 Report
Comments and Suggestions for Authors
Clarity and Accessibility:
Overall, your paper presents a comprehensive exploration of UAV logistics within the context of Urban Air Mobility. To enhance the accessibility of your work, focus on presenting complex concepts in a clear and reader-friendly manner. Consider providing brief explanations or definitions for technical terms to ensure that readers from various backgrounds can follow your arguments seamlessly.
Structural Organization:
The structure of your paper is well-defined with distinct chapters and sections. To further improve readability, consider begining each chapter or section with engaging introductions that outline the scope, purpose, and objectives. This provides readers with a clear roadmap of what to expect.
Contextualization:
While your introduction outlines the significance of UAV logistics and Urban Air Mobility, consider providing a broader context by briefly discussing the global trends and challenges in urban transportation that have paved the way for UAM. This can help readers understand the real-world context in which your research fits.
Application and Implications:
Throughout your paper, make an effort to connect your findings and analyses to real-world applications and implications. How might your research influence decision-making in urban logistics planning? Explicitly discussing the practical outcomes of your work can make your research more impactful.
Explanatory Captions:
When using graphs, figures, or charts, ensure that you provide clear and explanatory captions. Captions should succinctly convey the main point or finding illustrated by the visual, helping readers grasp its significance without needing to refer back to the main text.
Future Directions:
In your conclusion, when discussing future work and research directions, provide more specific details about the areas or aspects you intend to explore with the dynamic capacity assessment method. This can give readers a clearer sense of how your research might evolve and contribute to the field.
Glossary or Key Terms:
Given the technical nature of your paper, consider adding a glossary or a list of key terms at the beginning or end of your document. This can serve as a quick reference for readers encountering technical terminology.
Concluding Thoughts:
Your research on UAV logistics within the framework of Urban Air Mobility is valuable and timely. By focusing on clarity, contextualization, and practical implications, you can ensure that your work resonates with a wide range of readers and contributes meaningfully to the field. Keep up the great work!
The quality of English language in your manuscript is generally good. Your writing is coherent and conveys complex ideas effectively. However, there are some instances where clarity and grammatical precision could be further improved. Here are a few specific suggestions:
Sentence Structure and Clarity:
- In some sentences, the structure can become quite complex, making it challenging to follow the flow of your ideas. Consider breaking longer sentences into shorter ones to enhance clarity.
- Ensure that each sentence communicates a single clear idea. Avoid combining multiple concepts within a single sentence, as this can lead to confusion.
Verb Tense Consistency:
- Maintain consistent verb tenses throughout the manuscript. Switching between past, present, and future tense can make the text less cohesive.
Preposition Usage:
- Pay attention to prepositions to ensure their accurate use. They can significantly impact the meaning of sentences. For instance, consider whether "in" or "within" is more appropriate in context.
Article Usage (a, an, the):
- Use articles appropriately to specify or generalize nouns. Be sure to include "the" when referring to specific entities or concepts, and use "a" or "an" before general nouns.
Pronoun Reference:
- Ensure that pronouns (e.g., "it," "we") have clear antecedents. Readers should easily identify the noun to which the pronoun refers.
Subject-Verb Agreement:
- Maintain subject-verb agreement in your sentences. Plural subjects should correspond with plural verbs, and singular subjects with singular verbs.
Conjunctions and Linking Words:
- Utilize conjunctions (such as "and," "but," "however") and linking words to establish smooth transitions between sentences and paragraphs. They help guide readers through your ideas.
Parallel Structure:
- Ensure parallel structure in lists and comparisons. Items within a list or comparison should follow the same grammatical structure.
Overall, while the quality of English language is commendable, paying careful attention to these specific aspects can elevate the clarity and precision of your manuscript even further. It's a valuable step to ensure that your ideas are communicated effectively to a wide range of readers.
Reviewer 2 Report
Comments and Suggestions for Authors
The literature review can be improved, it's not clear what have done in the literature and what is the contribution. The importance of the model could be explained more. The formulas could be explained in more details. These are some comments about specific parts of the paper:
1. Define UAM and UAV in abstract in abstract.
2. In last paragraph of introduction, explain some of the airspace capacity assessment in the literature.
3. In last paragraph of introduction section, explain why is this: existing airspace capacity assessment methods are difficult to apply directly to low altitude airspace.
4. In the first paragraph of section 2.1. Problem Description explain what is operation mode of CaiBird post ?
5. In The First Objective section, revise the first sentences, it’s not clear what you’re trying to explain.
6. In the first paragraph of The First Objective section, what do you mean by per operation of time? How per operation of time is reflected in the formula?
7. In section of Second Objective, it’s not clear what are risk, noise and cost. Explain each of them and how they can be distinguished from each other in an operating system by UAVs?
8. Typo- is should be in in this sentence: …represents the distance of the segment Lij is the grid r.
9. How did you come up with formula (13)?
10. At the end of Methodologies section where the whole model is presented, specify what is the type of each variable, for example x is a binary variable.
11. In Algorithm section, first paragraph, why is this: Compared with the NGSA algorithm, the NGSA-II algorithm has lower complexity
12. Is Figure 4 from the literature? If yes, mention the reference.
13. The. Written NGSA-II Algorithm solving process looks very messy and hard to follow.
14. Use references for Table 2 Parameter setting
In overall some sentences are hard to understand, they need to be revised to be more fluent. Some sentences are too long to follow.
Reviewer 3 Report
Comments and Suggestions for Authors
The paper presents the results of the actual challenge due to designing an effective logistics UAV route network. The relevance of the study is beyond doubt. The article is well-structured and categorized as scientific. The presented results are interesting and needed for solving logistics issues along cyclical routes using UAV.
Several suggestions were made after the paper reading:
1) It is recommended to have a separate section for literature review to highlight the strengths and weaknesses of prior research on solving logistical problems with UAVs, particularly low-flying ones. This will clarify the motivation and gaps in current knowledge.”;
2) It is better to present the motivation, novelty of the study, and knowledge gaps in a separate paragraph.;
3) Why is square in formula (11)? Mathematically, it will be better to display this equation without square.;
4) In line 292, written about the stratification of all individuals. Please, explain what it means in more detail and how stratification was conducted.;
5) Please add a Discussion where obtained results could be compared with other models resolving such issues.
Finally, the research results have significant importance for the improvement of quality urban logistics during cargo delivery using low-altitude UAV.
Comments on the Quality of English Language
The English text is legible, but it would be improved with proofreading to ensure proper punctuation and writing style.
